# Temporal trends of cutaneo-mucous histoplasmosis in persons living with HIV in French Guiana: Early diagnosis defuses South American strain dermotropism

Sophie Morote[1], Mathieu Nacher [2,3,4]*, Romain Blaizot [1,3,5], Balthazar Ntab[6], Denis Blanchet[5,7], Kinan Drak Alsibai [8], Magalie Demar[4,5,7,9], Félix Djossou[4,7,9], Pierre Couppié[1,4], Antoine Adenis [2,3]

1 Service de Dermatologie-vénéréologie, Centre Hospitalier de Cayenne, Cayenne, French Guiana, 2 Centre d'Investigation Clinique Antilles Guyane, INSERM 1424, Centre Hospitalier de Cayenne, Cayenne, French Guiana, 3 COREVIH Guyane, Centre Hospitalier de Cayenne, Cayenne, French Guiana, 4 DFR Santé, Université de Guyane, Cayenne, French Guiana, 5 UMR TBIP, Université de Guyane, Cayenne, French Guiana, 6 Département d'Information Médicale, Centre Hospitalier de l'Ouest Guyanais, Saint Laurent du Maroni, French Guiana, 7 Laboratory, Centre Hospitalier de Cayenne, Cayenne, French Guiana, 8 Service d'Anatomopathologie, Centre Hospitalier de Cayenne, Cayenne, French Guiana, 9 Service des Maladies Infectieuses et Tropicales, Centre Hospitalier de Cayenne, Cayenne, French Guiana

* mathieu.nacher66@gmail.com

**Data Availability Statement:** The French regulatory authorities have not approved making

## Abstract

Histoplasmosis is the most frequent opportunistic infection and the first cause of mortality in HIV-infected patients in French Guiana and presumably in much of Latin America. Mucocutaneous lesions of histoplasmosis are considered as rare and late manifestations of the disease. It has been debated whether the greater proportion of cutaneo-mucous presentations in South America relative to the USA was the reflection of *Histoplasma* strains with increased dermotropism or simply delayed diagnosis and advanced immunosuppression. The objective of this study was to describe the clinical presentation, frequency, prognosis and temporal trends of cutaneomucous histoplasmosis in French Guiana. A retrospective study of patients with AIDS-related disseminated histoplasmosis followed in the three hospitals of French Guiana was performed between 1981 and 2014. Incident cases of histoplasmosis, proved by pathology and/or mycological examinations, were studied. Mucocutaneous histoplasmosis was confirmed by a positive cutaneous or mucosal biopsy. Mucocutaneous lesions were polymorphic. Ninety percent of patients were profoundly immunocompromised patients (CD4<50/mm3) and over 80% were not on antiretroviral treatment. The frequency of mucocutaneous forms and case fatality of disseminated histoplasmosis within one month of antifungal treatment significantly decreased over time (p<0,001). In this South American territory, diagnostic and therapeutic improvements have led to the quasi disappearance of cutaneous manifestations. There may be South American dermotropism in the laboratory but at the bedside early diagnosis seems to be the main parameter explaining the proportion of cutaneomucous presentations in South America relative to the USA.

this data publicly available. The data can be accessed by qualified researchers after getting prior approval from the 'comité national de l'informatique et des libertés'. For this, the research proposal must be submitted to the CNIL following the indication on their website (CNIL https://www.cnil.fr/en/home). Once the approval of the CNIL has been obtained by the researchers, the anonymized and encrypted database will be sent by the research organization of COREVIH Guyane (corevih@ch-cayenne.fr). If you have any questions, it is possible to contact Marilyne ABIVEN (director of the indirect right of access service department of the CNIL) at mabiven@cnil.fr.

**Funding:** The authors received no specific funding for this work.

**Competing interests:** The authors have declared that no competing interests exist.

## Author summary

Histoplasmosis is the most frequent opportunistic infection and the first cause of mortality in HIV-infected patients in French Guiana and presumably in much of Latin America. Mucocutaneous lesions of histoplasmosis are considered as rare and late manifestations of the disease. It has been debated whether the greater proportion of cutaneo-mucous presentations in South America relative to the USA was the reflection of *Histoplasma* strains with increased dermotropism or simply delayed diagnosis and advanced immunosuppression. The objective of this study was to describe the clinical presentation, frequency, prognosis and temporal trends of cutaneomucous histoplasmosis in French Guiana. A retrospective study of patients with AIDS-related disseminated histoplasmosis followed in the three hospitals of French Guiana was performed between 1981 and 2014. Incident cases of histoplasmosis, proved by pathology and/or mycological examinations, were studied. Mucocutaneous histoplasmosis was confirmed by a positive cutaneous or mucosal biopsy. Ninety percent of patients were profoundly immunocompromised patients (CD4<50/mm3) and over 80% were not on antiretroviral treatment. The frequency of mucocutaneous forms and case fatality of disseminated histoplasmosis within one month of antifungal treatment significantly decreased over time. Hence, in this South American territory, diagnostic and therapeutic improvements have led to the quasi-disappearance of cutaneous manifestations. There may be South American dermotropism in the laboratory but at the bedside early diagnosis seems to be the main parameter explaining the proportion of cutaneomucous presentations in South America relative to the USA.

## Introduction

In French Guiana, histoplasmosis is the most common AIDS-defining infection and cause of death [1,2]. This situation is presumably similar in many parts of Latin America, where the diseases often goes undiagnosed -mostly mistaken for tuberculosis- and untreated for lack of diagnosis. [3] Histoplasmosis is underdiagnosed in South America because the diagnosis often requires invasive explorations or locally unavailable techniques. Tuberculosis is the primary differential diagnosis of histoplasmosis. [4] The appearance of mucocutaneous lesions observed in histoplasmosis is an important element of clinical orientation between these two diseases. [5] The cutaneous-mucous manifestations of histoplasmosis vary both in morphology and distribution. Papules, nodules, ulcers, vesicles, or erosive, ulcerative cutaneous lesions, ulcers of the oral or nasal mucosa have been described. Lesions can be localized or diffuse, multiple or isolated. [6–9] This polymorphism makes it an unspecific clinical sign, especially as their spectrum morphological is shared with many infectious and non-infectious pathologies common to AIDS patients. [8] However, they constitute a site for very easily accessible and minimally invasive biopsies, allowing a rapid diagnosis for a potentially life-threatening illness that is difficult to diagnose. [5] Cutaneous-mucous forms of histoplasmosis are considered late forms of the disease, occurring at an advanced stage of immunosuppression. [8,10] Uncommon in the United States, they are very frequently described in Latin America. [11–19] This contrast between the two continents has led to search for a different disease phenotype between the strains of histoplasmosis [8,16,20,21]. Hence studies, have suggested that South American isolates were more dermotropic [21] but others suggested these cutaneous and mucous manifestations mostly reflected diagnostic delays. [5] The clinical comparison between different countries is difficult because disentangling between-country differences in

access to care and diagnosis, and genetically-based phenotypic diversity of histoplasmosis is a challenge given the great diversity in health systems and the importance of genetic variation of *Histoplasma* isolates in Latin America. [22,23] In contrast, looking at the historical perspective of clinical presentation allows to study its changes as access to diagnosis improved over time assuming that endemic *Histoplasma* isolates remained genetically similar. The main objective of this study was hence to describe the temporal trends of cutaneomucous forms of histoplasmosis among HIV-infected patients in French Guiana.

## Methods

### Ethics statement

This database was approved by the French regulatory authority, the National Commission Informatique et Libertés (CNIL) on November 27, 1991. All included patients received information and signed an informed consent. The main objective of the FHDH is to describe and study the morbidity and mortality of patients living with HIV in the AIDS stage. Regarding the 1992 Histoplasmosis and HIV anonymized database, it was also approved by the CNIL (n° JZU0048856X, 07/16/2010), and by the French National Institute of Health and Medical Research institutional review board (CEEI INSERM) (IRB0000388, FWA00005831 18/05/2010) and the Comité Consultatif pour le Traitement de l'Information pour la Recherche en Santé(CCTIRS) (N° 10.175bis, 10/06/2010).

### Study type

An observational retrospective multicenter study was carried out from January 1, 1981 to October 1, 2014.

### Study population

The target population was defined by anyone with co-infection with HIV and histoplasmosis included in the Histoplasmosis and HIV database in Guyana. The source population was represented by all patients living with known HIV and followed in one of the three hospitals in French Guiana. The criteria were:

- Age >18 years

- Confirmed HIV infection,

- First episode of histoplasmosis proven by direct mycological examination, culture mycological or histological examination (excluding PCR) performed on various samples (plasma, bronchoalveolar lavage, myelogram, digestive biopsies, skin biopsies. . .) according to EORTC / MSG criteria. [24]

Thus, patients with recurrent histoplasmosis, an unproven episode of histoplasmosis (effective empirical antifungal therapy) or diagnosis based solely on the positivity of the PCR was not retained.

### Judgment criteria

The primary endpoint was defined by the presence of *Histoplasma capsulatum*, confirmed by skin and / or oral or nasal mucosa on direct examination mycological and / or mycological culture and / or anatomopathological examination of tissues (EORTC / MSG criterion (40)). The prognostic endpoint was defined by mortality within 30 days after initiation of antifungal treatment.

### Study conduct

The Histoplasmosis and HIV database was created in 1992. It concerned incident cases of American Histoplasmosis in patients infected with HIV in the three hospital centers of French Guiana (Cayenne, Kourou, and Saint Laurent du Maroni). The epidemiological, clinical, para-clinical, immunovirological and therapeutic data until 10–2014 were collected on a standard-ized case record form and entered in the database. The data collected concerned any incident episode of histoplasmosis in a HIV infected patient, known or concomitantly discovered, hos-pitalized in one of the three hospital structures in Guyana, and consisted of socio-demo-graphic, clinical care and treatment, laboratory, imaging and 30 days survival after antifungal therapy initiation data.

### Statistical analysis

The statistical analysis was carried out using STATA software.

Frequencies were calculated for categorical variables. Mean, median, standard deviation, interquartile range, were calculated depending on the type and distribution of variables.

Standard hypothesis testing: Chi2 test or Fisher's exact test for categorical variables, and the trend Chi2 for ordinal variables were used; for quantitative variables Student's t-test or Rank-sum test were used depending on their normality. The statistical significance threshold was p <0.05.

### Ethical and regulatory aspects

All HIV-infected patients living with HIV followed in the three hospitals of French Guiana were included in the DMI-2 database, administered by the Regional Coordination for the fight against HIV (COREVIH). This database is part of the French Hospital Database on HIV (FHDH) which is a national cohort of patients living with HIV whose socio-demographic, clinical and biological data, and therapeutics have been prospectively included since January 1, 1992.

## Results

### Cutaneomucous lesions

Thirty-one cases of cutaneous and/or mucosal histoplasmosis were listed among 349 cases his-toplasmosis, between January 1, 1981 and October 1, 2014, in French Guiana. Among the 31 cases of cutaneo-mucous histoplasmosis observed during the study period, there were 15 skin lesions, 13 mucosal lesions and 3 cases with simultaneously cutaneous and mucous lesions. The skin lesions observed were mainly papules (N = 14, 78%). Three ulcers and one nodule were also described. These lesions were diffuse most of the time, more frequently localized on the face (14/18). Among mucosal lesions, ulcerative lesions were the most frequent (N = 10, 67%), sometimes both vegetative and ulcerative (N = 3, 20%). There were also two cases of pal-atal fistulas described. Mucosal lesions were predominantly oral (N = 10, 62%) affecting mainly the palate (7/16), and lips (N = 8, 50%). There was one case of nasal mucosal involvement.

### Epidemiologic features

The average age was 41 years (+/- 10 years). The sex ratio (M:F) was 2.9, half (16/31) of patients were of foreign origin, but with a median length of stay in French Guiana of 18 [7–32] years. Twenty nine patients were from Cayenne Hospital, and 1 was from Saint Laurent and 1 from Kourou hospital. The majority of patients had no antiretroviral therapy at baseline (25/31),

nor prophylactic treatment for opportunistic infections (29/31). About a third (11/31) had a history of opportunistic infection. Histoplasmosis was the AIDS-defining event in two thirds of the cases (21/31). The median CD4 count was 30 [15–42]/mm3 and the majority of cases were profoundly immunosuppressed (28/31) with a CD4 count below 50/mm3.

## Clinical and biological features

Fever was present in 83.9% of patients, and the WHO performance score was > 2 in 38.7% of cases. Half of the patients had admission respiratory signs and/or radiological pulmonary signs mostly an interstitial syndrome type. Digestive signs were found in almost a third of the cases. The average hemoglobin level was 9 +/- 2 g/dL with hemoglobin < 10 g/dL in 65% of patients, the median neutrophil count was 1.9 G/L [1,6–3], and for platelets of 181 G/L [81–235]. The median CRP elevation was 43 [26–87] mg/L. The majority of cases did not have kidney failure with a median creatinemia of 80.5 [79–106] μmol/L. All patients were undernourished with mean albumin level at 26.3 +/- 6.5 g/L. Liver function tests were moderately disturbed with medians of aspartate aminotransferase and alanine aminotransferase of 39 [29–73] IU/L and 24.5 [16–39] IU/L, respectively. The median gamma glutamyl transferase and alkaline phosphatase concentrations were 60.5 [37–105] IU/L, respectively 110 [76–147] IU/L. The median level of lactate dehydrogenase was 483.5 [241–1193] IU/L and was over twice the normal value in half of the cases. Ferritinemia and triglyceridemia were available for less than a third of patients, with medians 1001 [483–1441] μg/L and 1.36 [1.17–2.1] mmol/L, respectively.

## Diagnostic and therapeutic data

Histoplasmosis serology was only performed in four cases and was still negative. The majority of cases were diagnosed by direct mycological (93.5%) and / or pathological (83.9%) examination. Mycological cultures contributed to the diagnosis of certainty of histoplasmosis in 67.7% of the cases. Itraconazole was the initial antifungal therapy in 67.8% of cases, while 25.8% benefited from amphotericin B, of which only 2 cases in the form liposomal. One patient received first-line intravenous fluconazole and one received no initial therapy.

## Mortality at one month

There are 8 early deaths out of the 31 cases of mucocutaneous histoplasmosis listed, i.e. a one-month mortality rate of 25.8% among these forms over the entire period studied.

## Temporal trends of incidence and mortality

The first case of mucocutaneous histoplasmosis diagnosed in a patient living with HIV in French Guiana was observed in 1989. From then on to 1998, there were an average of 2 cases per year, mostly cutaneous. After 1998, the year when fungal culture started in Cayenne Hospital, mucocutaneus cases were in majority mucous with an average incidence of less than 1 case per year. Fig 1 shows the incidence of cutaneous mucosal histoplasmosis over three time periods from 1989 to 2014. The first period was before highly active antiretroviral therapy (HAART) (on average 21% of patients in the HIV cohort received on antiretroviral drug) and before fungal culture, the second and third periods are two-8 year intervals after highly active antiretroviral therapy, liposomal amphotericin B and fungal culture became available. From 1997 to 2005 on average 75% of patients in the HIV cohort received HAART, and after 2006 on average 89% of patients in the HIV cohort received HAART. Cutaneous mucosal histoplasmosis decreased over time (P<0.001). This decrease was driven by the decrease in the

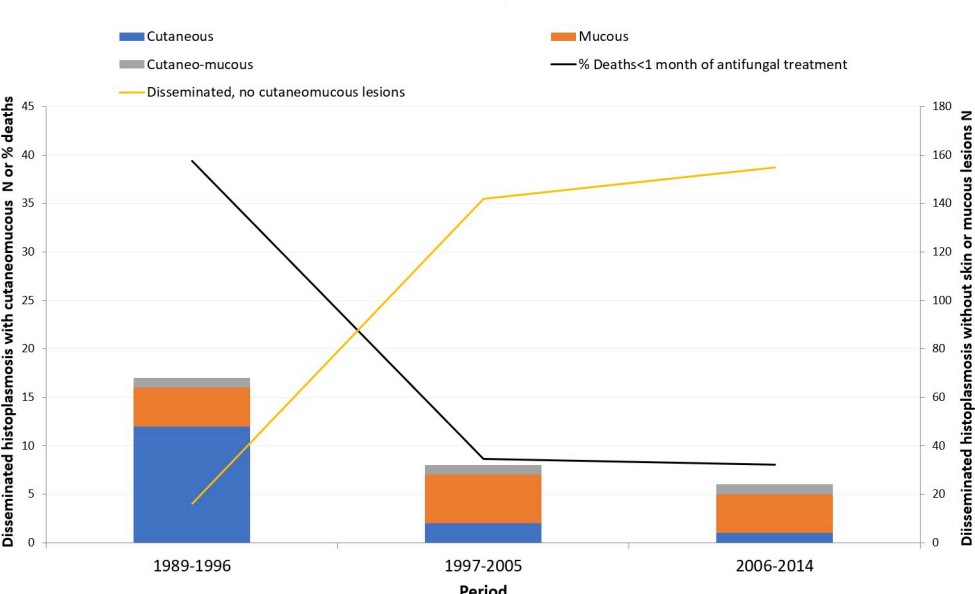

**Fig 1. Evolution of the number of disseminated histoplasmosis cases with and without cutaneo-mucous lesions among HIV-infected patients, French Guiana, 1989–2014.**

incidence of skin forms while the incidence of cases mucosal or mixed remains relatively stable over the three periods. During the same time the incidence of progressive disseminated histoplasmosis without cutaneomucous lesions greatly increased and case fatality rates within one month of antifungal treatment initiation declined fivefold among patients with cutaneomucous lesions (Fig 1).

Fig 2 describes the temporal evolution of the median CD4 count at diagnosis and the number of new HIV diagnoses. In the 1990s the spread of HIV led to a gradual increase in cases and, with time, a gradual decline and stabilization of the median CD4 count at the time of diagnosis.

## Discussion

The present results show that disseminated histoplasmosis with cutaneous and mucous lesions were present in equivalent parts, with a decrease in cutaneous lesions over time. In French Guiana, histoplasmosis diagnosis and awareness improved over time. The first diagnoses were made by dermatologists who saw patients with skin lesions, thereby showing that the disease was present. In this context, fungal culture was implemented, paraclinical investigations increasingly included biopsies and sample processing and pathological staining methods evolved. [25] The clinical lesion descriptions are in agreement with those found in the literature [7–9,26] from different geographical origins where different phylogenetic clades are observed. [23,27] The disease in United Stated is caused by *H. mississipiensis* and *H. ohiensis* while the disease in Latin America is caused by *H. capsulatum* and *H. suramericanum*. [28] The detailed phylogenetic description is rapidly gaining in precision, but there is a need for future prospective studies aiming to investigate the genetic background on the disease presentations. Although experimental data does suggest phenotypic differences in disease expression, notably dermotropism [16], what we observed at the "bedside" suggests that what was

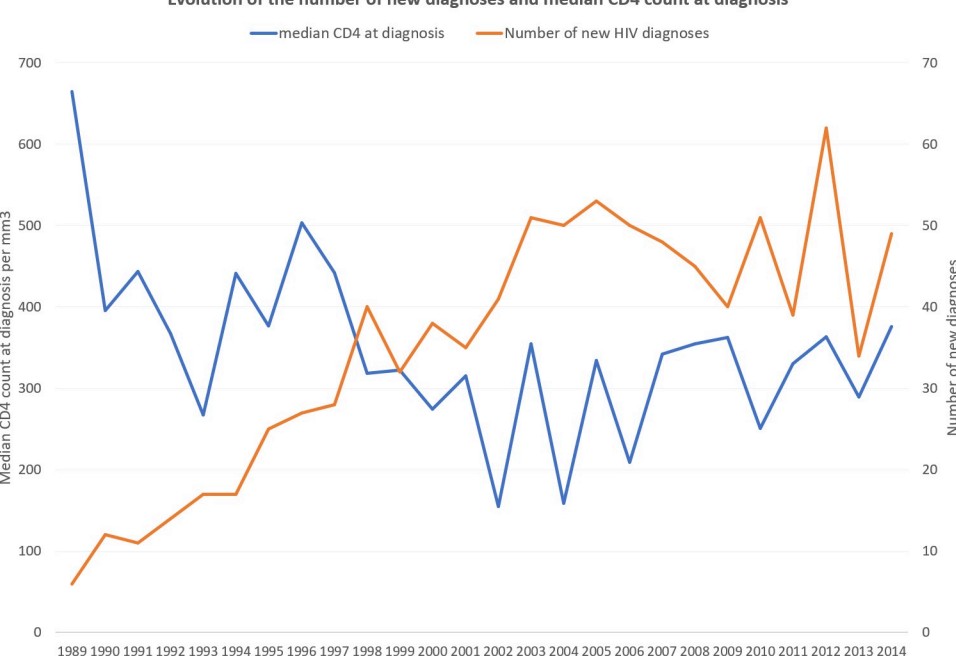

**Fig 2. Evolution of the number of new diagnoses and median CD4 count at diagnosis.**

observed "by the benchside" is less important than early diagnosis and treatment both of HIV infection, and disseminated histoplasmosis. Here, while assuming that the South American -dermotropic- *Histoplasma* strains remain similar in our endemic context, we show that with improvements in antiretroviral treatment and in histoplasmosis diagnosis have nearly suppressed cutaneous lesions. Hence, today, disseminated histoplasmosis in HIV-infected patients is clinically very similar to what is observed in the USA. [29] Although Basic science questions on virulence profiles are important, for clinicians and public health professionals access to HIV testing, and widespread availability of fungal diagnostic methods and treatments remains the most important things both to reduce advanced HIV disease [30] and the burden of disseminated histoplasmosis [31]. Here, 90% of patients with cutaneomucous lesions had CD4 counts below 50/mm3, and 80% were not receiving HAART. With improvements in testing and in the cascade of care [32] fewer patients reach levels of immunosuppression that allow dissemination of *Histoplama*. With progress in fungal diagnosis, antifungal treatment delays become rarer and dissemination is shortened, which is the most likely explanation of why cutaneous lesions disappeared. [33]

The present study has limitations that are linked to the retrospective nature of the data collection. Moreover, although there are many cases of histoplasmosis elsewhere, [34] the majority of cutaneous and mucous lesions were diagnosed in Cayenne, which may reflect a bias due to the hospitalization of patient in the dermatology department in Cayenne.

In conclusion, the present data suggest that in French Guiana clinical dermotropism is largely driven by late diagnosis as shown by the disappearance of such presentations with improvements in diagnosis and treatment. Whereas in the USA antigen detection allows rapid early diagnosis, in most of Latin America it is often late, which is a very plausible explanation for the greater proportion of cutaneous forms in Latin America relative to the USA. [21]

## Author Contributions

**Conceptualization:** Sophie Morote, Mathieu Nacher, Pierre Couppié, Antoine Adenis.

**Data curation:** Pierre Couppié, Antoine Adenis.

**Formal analysis:** Sophie Morote, Mathieu Nacher.

**Investigation:** Romain Blaizot, Denis Blanchet, Kinan Drak Alsibai, Magalie Demar, Félix Djossou, Pierre Couppié.

**Writing – original draft:** Mathieu Nacher.

**Writing – review & editing:** Sophie Morote, Romain Blaizot, Balthazar Ntab, Denis Blanchet, Kinan Drak Alsibai, Magalie Demar, Félix Djossou, Pierre Couppié, Antoine Adenis.

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
