## [Decision Letter · Decision Letter 0]

2 Jul 2020

Dear Pr. Nacher,

Thank you very much for submitting your manuscript "Temporal trends of cutaneo-mucous histoplasmosis in persons living with HIV in French Guiana: early diagnosis defuses South American strain dermotropism" for consideration at PLOS Neglected Tropical Diseases. As with all papers reviewed by the journal, your manuscript was reviewed by members of the editorial board and by several independent reviewers. The reviewers appreciated the attention to an important topic. Based on the reviews, we are likely to accept this manuscript for publication, providing that you modify the manuscript according to the review recommendations. 

Sincerely,

Joseph M. Vinetz

Deputy Editor

Joseph Vinetz

Deputy Editor

Reviewer's Responses to Questions

**Key Review Criteria Required for Acceptance?**

**Methods**

-Are the objectives of the study clearly articulated with a clear testable hypothesis stated?

-Is the study design appropriate to address the stated objectives?

-Is the population clearly described and appropriate for the hypothesis being tested?

-Is the sample size sufficient to ensure adequate power to address the hypothesis being tested?

-Were correct statistical analysis used to support conclusions?

-Are there concerns about ethical or regulatory requirements being met?

Reviewer #1: The methods are adequate despite the limitation of every retrospective study.

Reviewer #2: The study design does not seem adequate to the proposed objectives. Incidence cannot be estimated with a cross-sectional study. The authors demonstrate that they have a valuable database, but that it should be better explored. This imperfection is critical and, therefore, the whole text should be rewritten. The wrong choice of a study design ends up causing obstacles to the data analysis.

There is a lack of information regarding the clinical presentation of histoplasmosis, like measuring the difference between the onset of symptoms and the date of diagnosis. The same applies to findings regarding HIV infection. Information such as CD4 nadir, course of HIV infection, and year of HIV diagnosis. Understand that the diagnosis of an opportunistic infection may have a different interpretation if it occurred before or after the HAART policy introduction. It may reflect the high occurrence of late diagnoses, low adherence to HAART, or resistance to HAART in the sample. It is also imperative to analyze histoplasmosis incidence based on the prevalence of HIV infection in the pre and post HAART periods.

**Results**

-Does the analysis presented match the analysis plan?

-Are the results clearly and completely presented?

-Are the figures (Tables, Images) of sufficient quality for clarity?

Reviewer #1: The results clearly and completely presented

Reviewer #2: Data analysis is influenced due to the wrong choice of study design.

**Conclusions**

-Are the conclusions supported by the data presented?

-Are the limitations of analysis clearly described?

-Do the authors discuss how these data can be helpful to advance our understanding of the topic under study?

-Is public health relevance addressed?

Reviewer #1: Conclusions are adequate

Reviewer #2: Conclusions are influenced due to the wrong choice of study design.

**Editorial and Data Presentation Modifications?**

Reviewer #1: No

Reviewer #2: (No Response)

**Summary and General Comments**

Reviewer #1: The manuscript “Temporal trends of cutaneo-mucous histoplasmosis in persons living with HIV in French Guiana: early diagnosis defuses South American strain dermotropism” by Morote et al, investigates a deeply-discussed topic regarding the clinical manifestation of HIV-associated histoplasmosis between USA and Latin America. The manuscript and the retrospective study (despite the obvious limitations) are well presented and designed and should be considered for publication. There are few minor observations:

Introduction

Histoplasmosis – do not capitalize 

Histoplasma – Italicize

“In contrast, looking at the historical perspective of clinical presentation allows to vary access to diagnosis assuming” – Does not read well…please clarify “allows to vary access”

Methods

“all patients living with HIV known” – HIV-living patients?

Discussion

There is a lack of discussion regarding the genetic background of the players between North and Latin America: The disease in United Stated are caused by H. mississipiensis and H. ohiensis while the disease in Latin America is caused by H. capsulatum and H. suramericanum (See Sepulveda et al. MBio 2017). Prospective studies aiming to investigate the genetic background on the disease presentations are needed.

How this trend in of cutaneo-mucous histoplasmosis presentations is comparable to other fungal infections caused by Onygenales (i.e.: paracoccidioidomycosis and emmergomycosis) in HIV patients? Late diagnosis in other mycoses caused by dimorphic fungi can lead to skin disseminated disease?

Reviewer #2: Please, note that tuberculosis, with a microbiologically confirmed diagnosis, is definitely the most common opportunistic infection related to AIDS in Latin America. It cannot be valid in French Guiana but must be addressed in the text.

PLOS authors have the option to publish the peer review history of their article (what does this mean?). If published, this will include your full peer review and any attached files.

Reviewer #1: Yes: Marcus de Melo Teixeira

Reviewer #2: Yes: ALBERTO DOS SANTOS DE LEMOS
---

## [Editor Report · Decision Letter 1]

31 Jul 2020

Dear Pr. Nacher,

We are pleased to inform you that your manuscript 'Temporal trends of cutaneo-mucous histoplasmosis in persons living with HIV in French Guiana: early diagnosis defuses South American strain dermotropism' has been provisionally accepted for publication in PLOS Neglected Tropical Diseases.

Best regards,

Joseph M. Vinetz

Deputy Editor

Joseph Vinetz

Deputy Editor

---

## [Editor Report · Acceptance letter]

10 Oct 2020

Dear Pr. Nacher,

We are delighted to inform you that your manuscript, "Temporal trends of cutaneo-mucous histoplasmosis in persons living with HIV in French Guiana: early diagnosis defuses South American strain dermotropism," has been formally accepted for publication in PLOS Neglected Tropical Diseases.

Best regards,

Shaden Kamhawi

co-Editor-in-Chief

Paul Brindley

co-Editor-in-Chief
